# Evaluation of the Complementary Health Provision of the Podiatric Foot Care Program for Diabetic Patients in Catalonia (Spain)

**DOI:** 10.3390/ijerph18105093

**Published:** 2021-05-11

**Authors:** Jessica Ruiz-Toledo, Antonio J. Zalacain-Vicuña, Elena de Planell-Mas

**Affiliations:** Department of Clinical Sciences, Faculty of Medicine and Health Sciences, University of Barcelona, 08907 L’Hospitalet de Llobregat, Barcelona, Spain; azalacain@ub.edu (A.J.Z.-V.); elenaplanell@ub.edu (E.d.P.-M.)

**Keywords:** type 2 diabetes, foot ulcer, amputation, health care, collaborative practice, diabetes complications

## Abstract

The Catalan diabetic foot health program was established in 2009 in order to prevent complications caused by type 2 diabetes. This study aims to describe its application from 2009 to 2018. The objective was to describe diabetic foot care provision in the National Health System of Catalonia between 2009 and 2018, including the number of patients and professionals involved, the causes behind patients’ visits, and the most demanded codes for diagnosis and treatment filled by the podiatrist in each consultation during 2018–2020. This description was addressed through an analysis of the database provided by the Association of Podiatrists to evaluate the implementation of the program. The results for the diabetic foot health program in Catalonia showed a growth in demand from 2009 (1726) to 2018 (213,095) in terms of visits and from 2009 (1541) to 2018 (104,629) in terms of patients. The number of registered podiatrists from 2009 to 2018 increased from 165 to 470. The most commonly used diagnosis codes were (a) without sensory alterations in control and treatment of grade 1 lesions; (b) grade 0 without neuropathic, vascular, structural, or biomechanical alteration; (c) no sensory structural alterations in the foot; (d) keratopathies. The treatments most commonly used were (a) conservative (chiropody), (b) without ortho-podiatric treatment, and (c) plantar supports. The conclusions show that the health program is in great demand amongst the population. Similarly, the coding system has made it possible to identify the diagnosis and treatment of such demand.

## 1. Introduction

Diabetes Mellitus represents an economic cost for different countries [1]. Furthermore, it is the first cause of mortality due to cardiovascular complications [1]. The central problem of this disease is the dramatic world growth rate (in 2019, 463 million adults had diabetes [1]. The fact that glucose levels are uncontrolled increases cardiovascular complications; therefore, preventive treatments increase quality of life and reduce costs [1]. Planning national or global interventionist programs is one of the strategies of prevention health programs. In 2008, the “Strategy in Diabetes” was established in Spain through the creation of the Follow-up and Evaluation Committee, and it was assessed in 2010 [2]. In this assessment, several approaches were proposed to facilitate the implementation of tools and programs to diagnose and prevent Diabetes Mellitus, with the collaboration of Spain’s Autonomous Communities (regions). In Spain, the problem has been approached through different national measures in each region. These include the UDEN project—Girona Region (UDENTG), Andalusia’s Integral Plan of Diabetes (2009–2013), the Canary Islands’ Program for the Prevention and Control of Atherosclerotic Vascular Disease (specifically addressed at diabetic foot interventions (2000–2013)), and Extremadura’s 2014–2018 Integral Plan of Diabetes. In the case of Catalonia, a decree-law was created (28/2009) for podiatric attention to diabetic feet as a complementary benefit of the National Health System, which is described in this research paper [3].

Other international measures to approach this problem include the creation of the International Diabetes Federation’s Diabetic Foot International Work Group (IWGDF), which describes Toe and Flow, the Multidisciplinary Diabetic Foot Unit model from the USA; the “Foot Protection Service”, the Spanish DP-TRANSFERS (Catalonia); Epredice in Europe; and the multidisciplinary foot care service developed in the UK [4].

Interventions for risk factors can reduce amputations in cases of foot ulceration by up to 70% through health programs [5]. Strategies including prevention, education, and patient follow-up can reduce amputation rates by between 49% and 85% [6]. International guidelines recommend that it is essential that a podiatrist be included in the multidisciplinary diabetic foot care team [7].

“It is estimated that diabetes is associated with 11.3% of deaths to world level” [1]. There are 425 million people with diabetes in the world, and it is estimated that this will increase up to 700 million people by 2045 [1]. In Spain, the total prevalence of type 2 diabetes (DM2) was 6 to10% in 2017 [8]. In addition, “in the US, diabetic foot complications can represent up to 30% of excess medical costs for patients with diabetes” [9].

The etiology of diabetic foot ulcerations is complex; “50–70% of all lower limb amputations are due to diabetes” [10]. Ulcers are the leading cause of lower limb amputation, along with other factors such as diabetic peripheral neuropathy and peripheral vascular disease [11].

To address the management of the diabetic foot, the guidelines developed and the policy measures contemplated in the Berlin Declaration (2016) have been implemented. Foot prevention and care programs are cost-effective interventions for the health system.

About 15–25% of patients with diabetes develop a foot ulcer during their lifetime [11]. A total of 50% of ulcers are of neuro-ischemic origin, with 15% being exclusively ischemic [12]. These foot issues have a high financial cost. Thus, cost-effectiveness and cost-utility analyses are needed to prioritize and allow health management services to make the correct choices for approaching this prevalent chronic disease [13]. “These patients are between 10 and 24 times more likely to have an amputation than non-diabetics” [14]. Patients with previous amputations tend to have future amputations and therefore require more follow-up by multidisciplinary professionals [15]. ”The 5-year mortality rate exceeds 70% with a lower-extremity amputation” [16].

The screening of at-risk patients, in addition to a multidisciplinary approach involving podiatrists and primary care professionals, can help to prevent amputations and long-term complications and provide a platform for the implementation of prevention strategies. Between 49% and 87% of all foot problems in patients with diabetes are potentially avoidable [17].

In Catalonia in 2020, 7.9% of the population presented with diabetes, with greater prevalence in men than in women. Diabetes in Catalonia has increased since 1994 (4.8%), according to the 2020 Catalonia Health Survey (ESCA%) [18]. The Spanish public health system provides free primary and specialist care, although in Catalonia, podiatric care did not use to be covered. Via legislation passed in 2009, the Catalan health service established a public–private partnership to provide podiatric care for patients with diabetes with vascular disease and chronic neuropathy. The program aims to improve preventive measures and reduce risks by offering patients up to three free visits per year with a private podiatrist. Patients are referred through their corresponding primary care center [19].

The objective was to describe the diabetic foot program in the National Health System of Catalonia between 2009 and 2018, thus evaluating the number of patients and professionals involved, the causes behind patients’ visits, and the most demanded codes for diagnosis and treatment specified by the podiatrist for each consultation during 2018–2020.

Podiatrists are implementing guidelines for improvements in diabetic foot care, but it is necessary to continue working on compliance and standardization of prevention measures.

## 2. Materials and Methods

### 2.1. Study

The study involved a descriptive analysis of the program activities as well as the demand for program services.

### 2.2. Participants

A sample of n = 508,170 patients was obtained from the Catalan health system’s general database (2009–2018). All the patients analyzed presented a diagnosis of type 2 diabetes according to the following clinical criteria: (i) subjects with diabetes of over 10 years of evolution; (ii) subjects diagnosed with diseases associated with diabetes, such as neuropathy and vasculopathy; (iii) subjects who could present lesions on the foot such as helomas, oniychocryptosis, or neuropathic ulcers; and (iv) subjects with diabetes who needed orthopodological treatment. We also included subjects with diabetes who presented other physical problems, such as sight or mobility alterations [3].

Secondary data were provided by the registry of the Association of Podiatrists of Catalonia. The database contained the following information: number of visits, number of patients, and number of podiatrists for each of the years considered (2009 to 2018).

The diabetes foot commission of the Association of Podiatrists of Catalonia created a system of diagnosis and treatment codes (Table 1). These codes must be completed by the podiatrist for each patient (visit).

### 2.3. Procedure

The data obtained was collected by the Catalan health service through the invoicing of podiatric services declared by the Association of Podiatrists. Each visit under the aid plan at hand was registered in the common database, so that the data available constitutes the official data of this clinical activity. Therefore, the study of such distributions entails a comprehensive description of the clinical actions taken under this plan.

It is important to note that each podiatrist providing services to the health service must complete the list of codes and treatments. Podiatrists can specify as many codes as they consider appropriate.

### 2.4. Statistical Analysis

Microsoft Excel was used as a tool for descriptive statistical analysis. A descriptive analysis was performed for the data obtained, including the absolute number of podiatrists, the patients assisted, and the number of visits performed. An estimation of the slope of the longitudinal series (2009–2018) was made as well as an analysis of the most demanded codes (2018–2020). Data were presented in whole numbers and in percentages (%).

## 3. Results

Figure 1 shows the observed distribution of the variables registered during the analysis period. These results show a highly exaggerated increasing linearity in all three indicators. Complementarily, we obtained a total of 970,012 visits, 508,170 patients, and an increase in podiatrists providing services from 165 in 2009 to 470 in 2018. The three variables studied show an increase in the use of this supplementary benefit.

Regarding the Diagnosis and Treatment Code List filled out by the podiatrist for each activity (visit) during 2018–2020, the following results were obtained (Figure 2, Figure 3 and Figure 4). The most commonly used diagnosis codes were D15, D16, D0, and D7, and the most common treatment codes were T6, T4, and T1.

From the input of the code data record (June 2018–September 2020) there were 526,004 activities. Since the implementation in June 2018 of the diagnostic registry and treatment codes, 6001 codes were D19 (Grade 3: three alterations were present (neuropathic, vascular or small structural) out of 2,354,485 codes (0.25%). The presence and healing of an ulcer (D12) represents 10,461 codes of 2,354,485 codes (0.44%) in the three years of registration. The health program is also being widely accepted. Currently (2021), 689 podiatrists provide their services to the health program, with 213 new podiatrists joining the supplementary service. Conservative treatment (T6) by chiropody was the most frequently used by podiatrists: 519,327 codes from 2018–2020 of 1,090,424 treatment codes (47.62%) in three years. The proposed treatment (2018–2020) of plantar supports (T1) was applied to 3% of the patients visited.

## 4. Discussion

Catalonia (7,780,479 population) is the region in Spain with the third highest prevalence of diabetes, with figures similar to those of Israel, which has 9.19 million inhabitants. Several studies show that multidisciplinary treatment reduces amputations by 50% [20]. Primary health care professionals should be the first to detect, evaluate, and treat a patient at risk of diabetic foot syndrome. ”International guidelines recommend at least 3 levels of foot care treatment based on foot risk, and each level includes podiatrists” [16]. Blanchette et al. [21] compared wound closure time in individuals with diabetic foot ulcers (DFUs). Before the integration of a podiatrist, the ulcers were resolved in 44.6 weeks; after the integration of a podiatrist, wound closure time was reduced to 19.8 weeks. The authors suggest that a patient with DFUs who receives wound care from a multidisciplinary team including a podiatrist can improve both their healing rate and time.

Different professionals participate in the approach to the diabetic foot, including podiatrists as members of the Diabetic Foot Units. These units aim to reduce amputations as well as the number of hospital admissions and improve the quality of patients [22].

Early detection and treatment by the care team (diabetologist, podiatrist, orthopedic, rehabilitator) for digital and podiatric deformities in patients with diabetes play an important role in the prevention of ulcers and amputations [23]. 

Israel has been successful in reducing mortality due to cardiovascular diseases, but the burden of diabetes is relatively large. If we compare Catalonia with Israel, the latter implemented a National Diabetes Registry, which was established in 2014 by the Israel Center for Disease Control (National Program for Quality Indicators in Community Health Care (QICH)). This institution recorded a diabetes prevalence rate of 6.3%, similar to other Organisation for Economic Co-operation and Development countries. Nonetheless, it has the lowest hospitalization rate, since Israel promotes healthy lifestyles as well as comprehensive care [24].

The diabetic foot health program of Catalonia, established in 2009, has tried to prevent complications of type 2 diabetes in the foot. We consider that the results of the study can be representative of the program, as it was implemented in the same way and with the same protocols, and the assisted population is homogenous enough to elucidate the main problems. 

Diabetic ulcers imply a great financial cost for the health system [25]. The knowledge of clinically associated factors should be used in primary care and podiatry centers for the purposes of prevention. The Catalan program pursues this objective as well as the improvement of preventive measures and reduction of risks.

It has been shown that between 49 and 85% of foot problems can be prevented by appropriate measures and with the participation of different professionals, one of the key professionals is the podiatrist [26].

As has been previously stated, this work deals mainly with the diabetic foot through an evaluation of associated healthcare services, which have seen great demand since their inception. However, early detection of diabetes complications could help reduce increasing treatment costs.

Therefore, any program needs to be evaluated and improved, with information provided to assess return on investment. 

## 5. Conclusions

According to the results found, we conclude that the diabetic foot health program implemented in Catalonia in 2009 has followed a linear, progressive and increasing trend in the number of registered patients, visits and podiatrists. The most commonly used diagnostic codes were D15 (without alteration in control and treatment of grade 1 lesions), D16 (Grade 0: without neuropathic, vascular or structural and biomechanical alteration of the foot), D0 (no sensory alterations in the structure of the foot) and D7 (keratopathies), and the most common treatment codes were T6 (conservative), T4 (without treatment ortho-podiatric), and T1 (plantar supports). This program can be extrapolated to other regions/countries, thus helping to prevent complications from diabetes. In Spain, Andalusia introduced a podiatric care model similar to the Catalan model in 2019.

## Figures and Tables

**Figure 1 ijerph-18-05093-f001:**
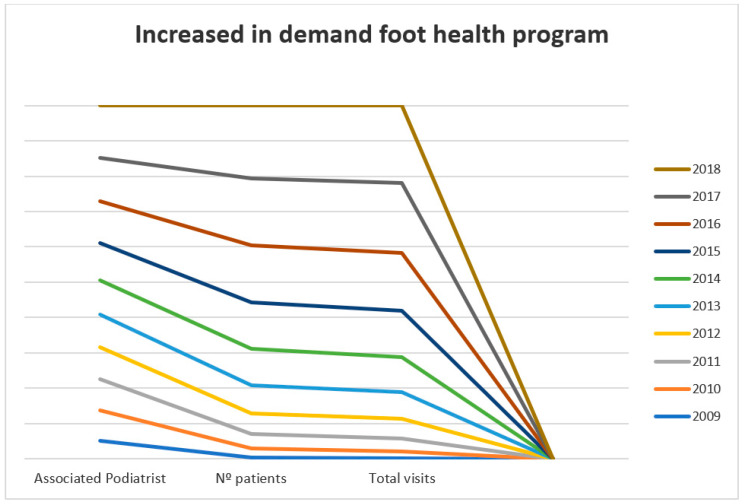
Distribution of total visits, number of patients and associated podiatrist (2009–2018).

**Figure 2 ijerph-18-05093-f002:**
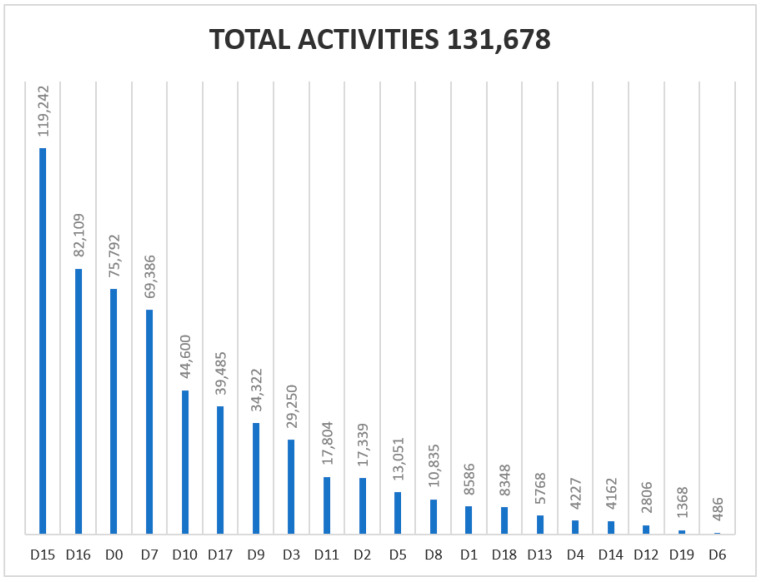
Graphic showing the usage of codes in 2018 (from June to December).

**Figure 3 ijerph-18-05093-f003:**
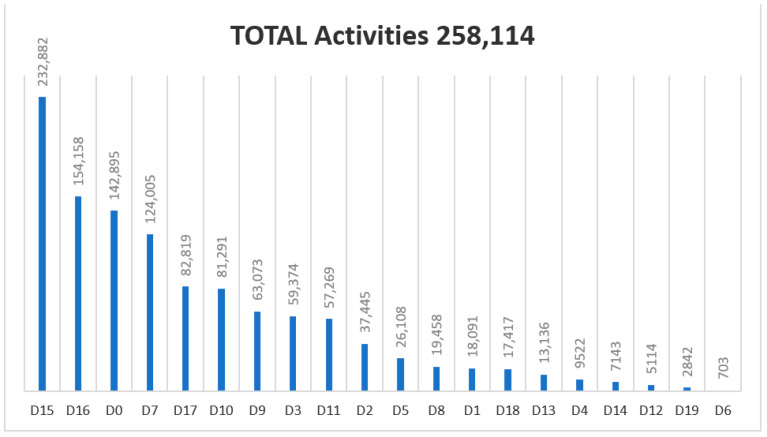
Graphic showing the usage of codes in 2019 (from January to December).

**Figure 4 ijerph-18-05093-f004:**
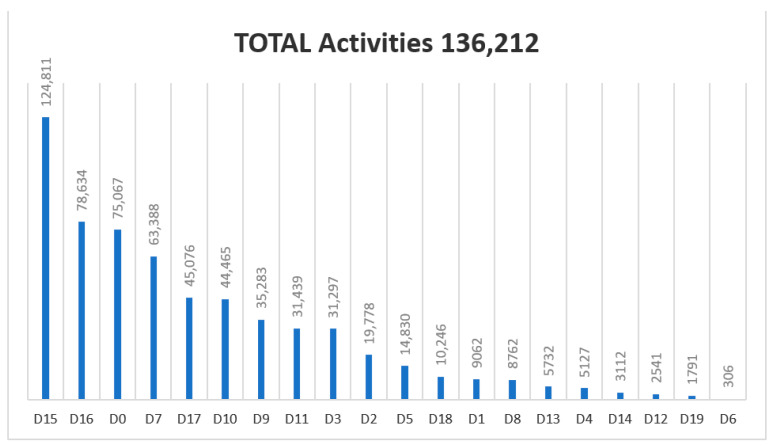
Graphic showing the usage of codes in 2020 (from January to September).

**Table 1 ijerph-18-05093-t001:** List of diagnosis and treatment codes filled in by the podiatrist for each activity.

List of Diagnosis Codes
**1—Diagnosis (Assessment of structural alterations in the foot)**
D0_ No sensory alterations
D1_Flat foot
D2_ Foot dig
D3_ Digital deformity
D4_ Hindfoot alterations
D5_ Biomechanical alteration
**2—Control and treatment of keratotic and nail lesions**
D6—Wart
D7—Keratopathies
D8—Fang infections
D9—Onychocryptosis
D10—Onychogryphosis
D11—Sensory alteration
**3—Control and treatment of grade 1 lesions (superficial) ITB index> 0.7**
D12—Presence and healing of ulcer
D13—Heel cracks
D14—Erosions and vesicles
D15—Without sensory alteration
**4.Diabetic foot assessment (neuropathic and vascular)**
D16—Grade 0: Without neuropathic, vascular or structural and biomechanical alteration of the foot
D17—Grade 1: One present alteration (neuropathic, vascular or structural of the foot)
D18—Grade 2: Two alterations (neuropathic, vascular or small structural)
D19—Grade 3: Three alterations present (neuropathic, vascular or small structural)
**List of Treatment codes**
**1.Proposed ortho-podiatric**
T1—Plantar supports
T2—Silicone orthosis
T3—Provisional descriptions
T4—Without treatment
T5—Prosthesis
**2.Control and treatment of keratotic and nail lesions**
T6—Conservative TTO (Podiatry)
T7—Pharmacological
TTO
T8—Surgical TTO
T9—Referral to a specialist

## Data Availability

Not applicable.

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
