# Peer review of "Evaluation of the Complementary Health Provision of the Podiatric Foot Care Program for Diabetic Patients in Catalonia (Spain)"

_ijerph, 2021, doi:10.3390/ijerph18105093_

Round 1

Reviewer 1 Report

I appreciate the opportunity to review this article.
I congratulate the authors for the article, because the theme is very interesting for today. Diabetes is present worldwide and is considered a worldwide public health problem.
Please remove the health descriptors from the Mesh controlled dictionary (https://www.ncbi.nlm.nih.gov/mesh/).
In the introduction: Please insert the references in the first paragraph (lines 44 to 58).
Please arrange the last paragraph of the introduction, where there is twice the word "objective" (line 107).
In the method section on "statistical analysis" please also insert that the results were presented in whole numbers and in percent (%) (lines 147 to 150).
I suggest reviewing the number of figures, which are currently 7. I suggest condensing figures 2,3 and 4 into a single graph, using a color or texture for each topic shown.
In the conclusions, please replace the diagnostic codes were D15 D16 D0 D7 and treatment codes T6 T4 T1 with the names (described in figure 1).
Congratulations to the authors. Success

Author Response

Thank you for your suggestions. Please find attached document to the reviewer.

Reviewer 2 Report

This is a manuscript talking about type 2 diabetes and foot provision relationship. This is a case collection study. I am very interested in your investigation. However, no external funding, all data from health system’s general database. This is a reviewer article, not research. From government data not from survey by themselves. Because that is not results. All figures and table. So I am not agree to accept this manuscript. 

Author Response

(The authors gave the same response as above.)

Reviewer 3 Report

The authors presented a study that explain the evaluation of the complementary health provision of the podiatric foot care program for diabetic patients in Catalonia.

To this review, the topic is interesting however there are several major limitations. The writing is also rather "rough", it's to improve the form.

Abstract: it's well written, the objective is clear.

Introduction: The authors need to include more literature, from line 39 to 58, only one paper is cited.

The authors need to improve the form, with more attention to punctuation.

Materials and Methods: To help the readers, the authors need to add a table with age of subjects, years of disease, how many subjects have diseases associated with diabetes (neuropathy/vasculopathy) and also a table/figure that summarises the lesions on the foot (line 126 - 127).

To help the readers, the authors should rearrange the figure 1 in two tables, one for the diagnosis and another one for the treatment.

At line 146, the title of the paragraph is "Statistical Analysis", but the authors don't perform any statistical test.

Results: It's not clear why in Figures 5,6,7, Diagnosis and Treatments are presented in the same chart, so to help the readers, according to the "Materials and Methods", the authors should arrange the Figures 5,6,7, dividing the Diagnosis and Treatment. 

Lines 184-185: the authors state that "Since the implementation in 2018 of the diagnosis and treatment codes, 6001 Grade 3 cases have been diagnosed", Grade 3 is referred to D19 (Figure 1), but in the figures 5,6,7, D19 is never reported.

Discussion: It's well written and clear. 

Conclusion: it's well written and clear and support the results.

Author Response

(The authors gave the same response as above.)

Reviewer 4 Report

Results section only include the result that can be summarized directly from the data the authors obtained. More detailed data analysis need to be conducted in order for this manuscript to be re-considered for publication.

Specifically, Figs. 5-7 are based on different time-line, where Fig. 5 is based on Jun-Dec of 2018, while Figs. 6-7 are based on Jan-Dec of 2019 and 2020. Also Figs. 5-7 only indicate the total number of activities on different codes. What about the trend of the activities w.r.t. the time-line? Is the number of activities for any of these codes increase or decrease from 2018 to 2020? 

Again, the results section in its current form only shows the apparent information that anyone could summarize from the data. Much more detailed data analysis need to be conducted.

Author Response

(The authors gave the same response as above.)

Round 2

Reviewer 2 Report

Dear Authors:

Thank you for responding to my recommendation.

The other reviewers are like your study. I saw them say.

Ok. All data are from The Catalan health system.

I can agree.

Reviewer 3 Report

The manuscript has been updated with all the requested revisions.

Thank to the authors, in my opinion the Manuscript is ready for the publication.

Good luck!